# Assessment of Magnetic Nanomaterials for Municipality Wastewater Treatment Using Biochemical Methane Potential (BMP) Tests

**DOI:** 10.3390/ijerph19169805

**Published:** 2022-08-09

**Authors:** Gloria Amo-Duodu, Emmanuel Kweinor Tetteh, Sudesh Rathilal, Martha Noro Chollom

**Affiliations:** Green Engineering Research Group, Department of Chemical Engineering, Faculty of Engineering and the Built Environment, Durban University of Technology, Durban 4001, South Africa

**Keywords:** anaerobic digestion, biochemical methane potential, chemical oxygen demand, nanomaterials, kinetics

## Abstract

Wastewater as a substrate potential for producing renewable energy in the form of biogas is gaining global attention. Herein, nanomaterials can be utilised as a nutrient source for microorganisms for anaerobic digestion activity. Therefore, this study explored the impact of seven different magnetic nanomaterials (MNMs) on the anaerobic digestion of wastewater via biochemical methane potential (BMP) tests for biogas production. The BMP assay was carried out with eight bioreactors, where each was charged with 50% wastewater and 30% activated sludge, leaving a headspace of 20%. Aside the control bioreactor, the other seven (7) bioreactors were dosed with 1.5 g of MNMs. This was operated under anaerobic conditions at a mesophilic temperature of 35 °C for 31 days. At the degree of 80% degradation of contaminants, the results that showed bioreactors charged with 1.5 g MNMs of TiO_2_ photocatalyst composites were more effective than those constituting metallic composites, whereas the control achieved 65% degradation. Additionally, the bioreactor with magnetite (Fe_3_O_4_) produced the highest cumulative biogas of 1172 mL/day. Kinetically, the modified Gompertz model favoured the cumulative biogas data obtained with a significant regression coefficient (R^2^) close to one.

## 1. Introduction

The development of energy-efficient centralised wastewater treatment systems to mitigate emerging pollutants and environmental challenges associated with the water sector is gaining global attention [1,2]. Meanwhile, there is a significant risk of freshwater and energy resource depletion because of increasing population growth and industrialization, as well as anthropogenic CO_2_ emissions [1,2,3].

South Africa, a country known for its scarcity of water, is also faced with high-strength organic chemical pollutants emitted by industries such as pharmaceuticals, petrochemicals, agrochemicals, mining, textiles, pulp and paper, and so on, posing threats to water bodies [4,5,6]. These recalcitrant pollutants typically enter the aquatic medium via industrial effluent that does not meet the discharge standards due to ineffective municipal Wastewater Treatment Plants (WWTPs) [5,7]. Water-soluble substances, in general, are easier to distribute and transport in the water cycle, and their direct impact on the ecosystem can be seen in a short period of time [6,8].

Furthermore, advancements in scientific environmental assessments reveal that recalcitrant contaminants (such as antibiotics) can still be detected in wastewater streams after a long period of time [9]. As a result, conventional WWTPs are incapable of dealing with high-concentration organics and emerging contaminants (ECs) such as pharmaceuticals (antibiotics), biomolecules (COVID-19 RNA), personal care products, food additives, and customized nanomaterials [6,8]. This has piqued the interest of many water sector stakeholders in improving the efficacy of WWTPs.

To comply with stringent bylaws, the energy required for the operation of conventional wastewater treatment plants (WWTPs) in conjunction with disposal and distribution systems is costly (South African-German Energy Programme—GIZ-SAGEN, 2015) [4,10,11]. As a result, using wastewater treated residue as an energy resource for biogas production to offset the WWTP’s energy consumption becomes interesting area for researchers [10,12]. Herein, anaerobic digestion (AD) has been one of the global technologies used for the degradation of high organic content of wastewater into biogas [12].

AD, on the other hand, involves the hydrolysis of complex organics to soluble and degradable molecules, as well as acidogenesis, acetogenesis, and methanogenesis [13,14]. Consequentially, the AD processes, which are predominantly employed as biological processes for industrial and municipal wastewater treatment, have their own challenges [15,16]. This is because most of them either operate at a very low capacity due to the numerous challenges posed by emerging and biorecalcitrant compounds and the rest have completely collapsed [17,18,19]. Thus, this issue must be addressed to improve water quality and biogas production. Several studies, which include reactor adjustment, addition of nanomaterials, co-digestion, etc., have been adopted over the years to curb this challenge [18,20].

Nanomaterials have exceptional size-dependent properties (10–150 nm), making them indispensable and superior for a wide range of applications. Examples include chitosan, titanium dioxides (TiO_2_), iron oxides, zinc oxides, zeolites, carbon nanotubes, copper oxides, and so on [21,22,23]. Furthermore, the presence of most of these chemical additives (iron- and aluminium-based coagulants) alters the chemistry of the sludge, resulting in complex organic contents. As a result, reducing sludge production while increasing caloric value via biological treatment in conjunction with NPs is a possibility [22,24,25,26]. A study by Ajay, et al. [20] reports that these metal NPs (iron, cobalt, nickel etc.) are inorganic additives that serve as micronutrients to the microorganisms in the AD to enhance methane and biogas production.

In this vein, the current study was conducted, where the application of magnetic nanomaterials for wastewater treatment using the biochemical methane potential test was evaluated to ascertain the effects on contaminant removals, the methane and biogas yield.

## 2. Materials and Methods

The wastewater and sludge were obtained from the eThekwini municipal wastewater treatment plant (Umbilo) in South Africa’s KwaZulu-Natal province of which the samplings of the wastewater and sludge were performed at the biofiltration system (BS) of the plant. The wastewater and sludge were characterized in accordance with water and wastewater examination standard methods [27]. The results obtained are shown in Table 1. The nanomaterials used in this study was synthesized using the co-precipitation synthesis method, which has been detailed in studies by Tetteh, et al. [28], Amo-Duodu, et al. [29], and Amo-Duodu, et al. [30]. The characteristics of the nanomaterials have been reported in these studies. The selection of these nanomaterials was conducted based on a study by Amo-Duodu, et al. [31].

### 2.1. Biochemical Methane Potential (BMP) Test

The BMP test was performed in accordance with the protocol reported by Jingura and Kamusoko [13] and Hülsemann, et al. [14] to attest to the effectiveness of the synthesized magnetic nanomaterials (MNMs) used for the study. This was completed using 8 Duran Schott bottles (1 L bioreactors) with air-tight caps and three outlets on each cap, which were placed in a thermostatic water bath (Figure 1). Table 2 presents the wastewater, activated sludge, and MNPs load distribution for each bioreactor. After charging the bioreactors (A–H), they were purged with nitrogen gas for 2 min and allowed to stand for two days to create an anaerobic environment. The bioreactor systems (A–H) were then run at a temperature of 35 °C for 30 days. The downward displacement technique was used to monitor the daily amount of biogas produced.

### 2.2. Water Quality Analysis

At the end of the 30-day digestion period, samples of the supernatant were taken and analysed from each setup. The remaining content was decanted from each setup, leaving the sludge behind for analysis. Before analysing, 5 mL of supernatant liquid was measured and diluted with distilled water using a dilution factor of 10. The reactor efficiency was calculated by estimating the contaminants removal (Equation (1)). The first-order and modified Gompertz models were used to determine the degree of degradation and stability of the biological system as a function of the cumulative biogas data obtained for a given run.
(1)$$Reactor\;efficiency=\left(\frac{C_{i}-C_{f}}{C_{i}}\right)\times 100$$
where, $C_{i}$ = Substrate influent and $C_{f}$ = Substrate effluent.

Additionally, the cumulative biogas data obtained was fitted on a modified Gompertz model and first-order kinetics model as expressed in (2) and (3), respectively. This was used to estimate the biogas yield.
(2)$$Y\left(t\right)=Ym.\exp\left(-\exp\left[\frac{2.7183R_{max.}}{Ym}\left[\lambda -t\right]\right]+1\right)\;$$
(3)Y(t)=Ym [1−exp(−kt)] 
where *Y(t)* = Cumulative methane yield (mL/g COD), *Ym* = maximum methane yield (mL/g COD), *k* = rate constant (1/day), *R_max_* = maximum methane production rate (mL/g COD.day), *k = (R_max_.e/Cm)* = maximum specific substrate uptake rate per the maximum biogas production (1/day), *ʎ* = Lag phase (day), and *t* = time (day).

## 3. Results and Discussion

The biostimulation effect and treatability efficiency of MNMs (Table 2) by each bioreactor (setup A–G), compared with the control system (setup H), is presented in this section. This was based on the BMP results obtained.

### 3.1. Effect of MNMs on Contaminants Removal from BS Wastewater

The BMP test’s treatability performance was evaluated, and the contaminants’ removal (%COD, %colour, %turbidity) from the BS wastewater sample was evaluated for each bioreactor. From the findings of the study, magnetised photocatalyst (TmF) had a high significance in the removal of specific contaminants in wastewater samples (Figure 2) of a removal efficiency of COD, colour, and turbidity, which was found to be 91.60%, 78.95%, and 97.81%, respectively.

Figure 2 depicts the outcome of evaluating MNMs for the treatment of BS wastewater, where setup D with MNM (TmF) performed admirably with approximately 79% colour removal. This could be because the additive MNM (TmF) is composed of TiO_2_ and Fe, which are trivalent ions with high oxidation-reduction properties capable of oxidizing a wide range of organic pollutants [32,33,34]. In this case, the overall performance of the MNM-containing bioreactors was found to be preferable and superior to that of the control bioreactor H (no MNM). The order of COD degradation was found to be F (93.70%) > A (92.59%) > B (91.90%) > D (91.60%) > E (90.76%) > C (88.07%) > G (79.83%) > H (54.96%).

The colour and turbidity removal performance of the bioreactors with MNMs was compared to the control (setup H). It was observed that the control has a removal of 45.61% and 60.79% for colour and turbidity removal, respectively, as shown in Table 3 and Figure 2. The removal efficiency of colour for the bioreactors with MNMs dosed had above 60% and a turbidity removal above 80%, as presented in Table 3. The findings also suggested that bioreactors with MNMs composed of TiO_2_ photocatalyst (ChTmF, ATmF, TmF, and CTmF) had good biodegradability of the contaminant, which could be attributed to their high sorption ability for high-strength organic contaminants [35]

### 3.2. Biogas and Methane Yield of BMP System for BS Wastewater

Figure 3 depicts the cumulative biogas production of reactors A-H. Bioreactor A, dosed with magnetite (mF) additives, produced 1172 mL/day of biogas, which was almost double that compared to control reactor H (525 mL/day). This result was in contrast to reactor D with MNM (TmF), which was very effective for water quality improvement efficiency for colour and turbidity removal, as reported by other studies [22,36,37]. TmF is a well-known photocatalyst whose surface is more active and gets excited when exposed to UV-light [38]. This excitation causes the release of electrons and holes radicals, which help in the adsorption of contaminants and reduction of CO_2_ to methane in the presence of hydron ions [38,39,40]. However, in this study the TmF was not exposed to any UV-light; hence, it was not as effective as it could have been, and this could be the reason for the above observation of its low biogas yield. Similarly, the increase in biogas production validates previous reports because MNMs produce radical ions that act as a reducing agent during methanogenesis activity, as shown in Equation (4) [15,18,41].
(4)$$CO_{2}+4H_{2}\;\to CH_{4}+2H_{2}O\;\;\;\;\;\;\;\;\;$$

Figure 4 depicts the methane yield for the BMP setups A-H. The findings of this study support previous research on the use of MNMs and other trace metal solutions such as Fe, Cu, Ni, Zn, Ti, and Mg for biogas generation and methane enhancement [42,43,44]. Liang, et al. [43] reported on the usage of iron-based nanomaterials to enhance methane and biogas generation. Aside from biogas production, the MNM additions improved the methanation mechanism. This resulted in a significant percentage (>80%) of methane composition as compared to the control setup H (Figure 4). Thus, setups A, B, C, and D recorded >90% methane composition, but the control (setup H) had a methane output of 65%. Importantly, the increase in methane content is very efficient in terms of heat and power consumption.

### 3.3. Kinetic Study of the BMP System

Kinetic modelling was used as an acceptable method to determine the kinetic conditions of the bioreactors as a function of the biogas produced. This was carried out to obtain information about the reactor’s kinetic degradation to avoid impending AD reactor failure due to poor operation [45]. The obtained cumulative biogas data was fitted using the modified Gompertz and first-order models to ascertain the substrate–microbe utilization for the biogas production [18,45]. Table 4 and Figure 5 show that the systems fitted better on the modified Gompertz models with R^2^ values greater than 0.98. Clearly, the results (Figure 3) indicate that the presence of MNM additives accelerated the kinetics degradation activity, which increased the biogas production [45]. As a result, the models’ predicted biogas values were relatively higher than the measured biogas values (Yt). In essence, the BS wastewater stream data from the bioreactors with MNM additives favoured the modified Gompertz kinetic model, with their lag of phase (ʎ) being within 4–9 days, attesting to the reactor’s rapid response [45]. Moreover, the minimum sum of squares errors (SSE) denotes the models’ statistical significance (*p* > 0.05) and predictability. Likewise, it may provide knowledge on how to design an industrial-scale reactor operating under similar conditions to be viable with MNM additives.

## 4. Conclusions

The magnetic nanomaterials (MNMs) investigated demonstrated having potential to enhance the AD process methanogens’ rapid response for biogas production, reducing sludge production and improving the wastewater treatment quality. A biochemical methane (BMP) test was used in this study to evaluate the MNMs’ biostimulation effect on anaerobic digestion of wastewater for biogas production and treated wastewater for reuse. This also enhanced the kinetic stability of the AD system and improvement of the biogas produced. MNMs composed of TiO_2_ photocatalyst composites (ATmF, TmF, CTmF, and ChTmF) were found to be more effective than those composed of metallic composites. Furthermore, the degree of degradation with BMP setups charged with MNMs demonstrated 70–80 percent removal of the COD, colour, and turbidity when compared to the control system, which achieved 50–65 percent efficiency without any MNM additives. It is found that bioreactor with TmF additives demonstrated a critical pathway for converting wastewater into circular-economy resources (energy).

## Figures and Tables

**Figure 1 ijerph-19-09805-f001:**
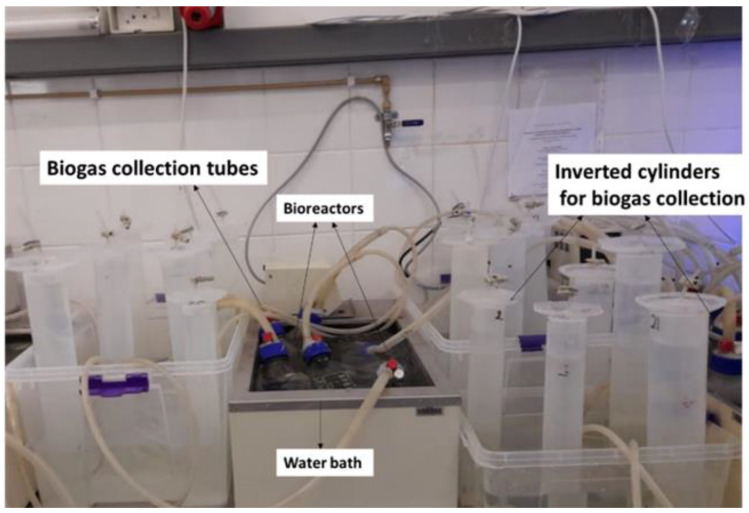
The schematic diagram of the biochemical methane potential (BMP) test setup.

**Figure 2 ijerph-19-09805-f002:**
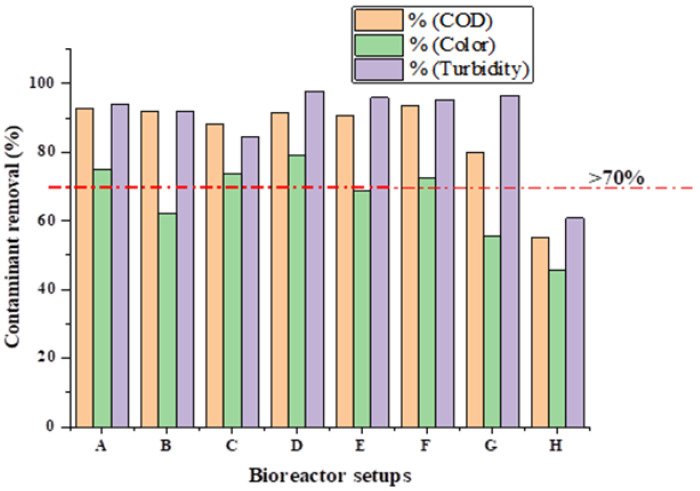
Contaminant removal for bioreactors A (mF), B (NmF), C (CmF), D (TmF), E (ChTmF), F (CTmF), G (ATmF), and H (no MNM) of MNM loading of 1.5 g at 35 °C for 30 HRT.

**Figure 3 ijerph-19-09805-f003:**
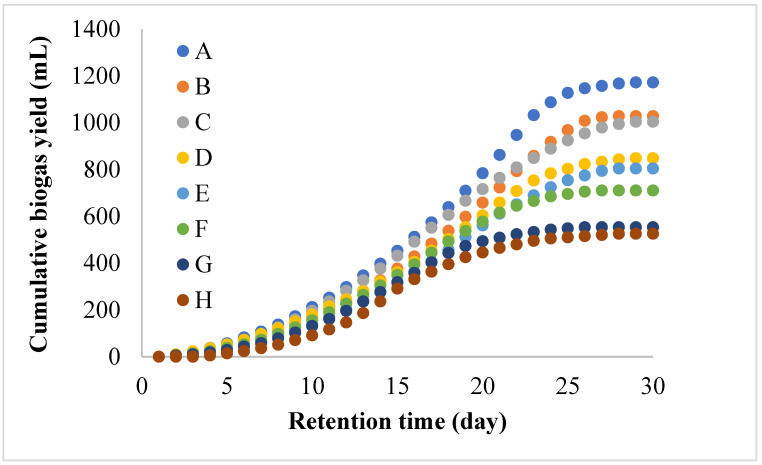
Cumulative biogas yield for bioreactors A (mF), B (NmF), C (CmF), D (TmF), E (ChTmF), F (CTmF), G (ATmF), and H (no MNM) of MNM loading of 1.5 g at 35 °C for 30 HRT.

**Figure 4 ijerph-19-09805-f004:**
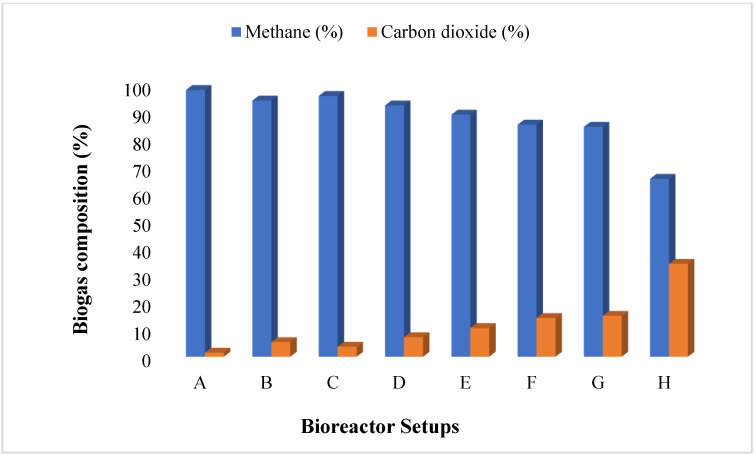
Biogas composition for bioreactors A (mF), B (NmF), C (CmF), D (TmF), E (ChTmF), F (CTmF), G (ATmF), and H (no MNM) of MNM loading of 1.5 g at 35 °C for 30 HRT.

**Figure 5 ijerph-19-09805-f005:**
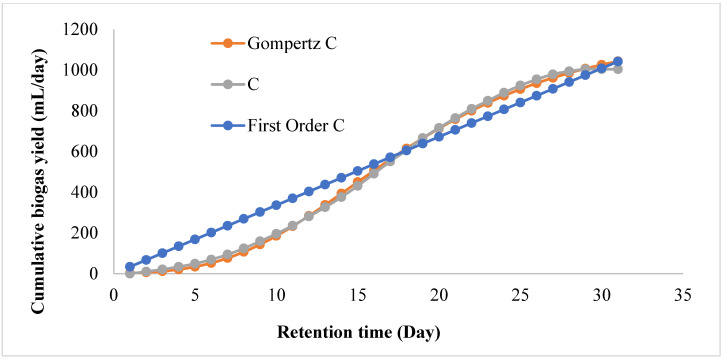
Fitting of cumulative biogas yield of bioreactor C (CmF) with highest R^2^ (0.9986) on first-order and modified Gompertz kinetic models.

**Table 1 ijerph-19-09805-t001:** Characterization of wastewater and activated sludge samples.

Wastewater	
Parameters	Biofiltration System (BS)
Chemical oxygen demand (COD) (mg/L)	2380 ± 32
Colour (465 nm, Pt.Co)	570 ± 7.6
Turbidity (NTU)	73.2 ± 12.5
pH	7.42 ± 3.6
Activated sludge	
Total Solids (TS) (mg TS/L)	304.5 ± 23.6
Volatile Solid (VS) (mg VS/L)	229.5 ± 2.65
VS/TS (%)	75.37 ± 3.5

**Table 2 ijerph-19-09805-t002:** Experimental matrix for BMP test.

Setup	MNPs Loading (g)	Symbol (s)	Wastewater (L)	Sludge (L)
A	1.5 Fe_3_O_4_	mF	0.5	0.3
B	1.5 NiFe_2_O_4_	NmF	0.5	0.3
C	1.5 CuFe_2_O_4_	CmF	0.5	0.3
D	1.5 TiO_2_Fe_2_O_4_	TmF	0.5	0.3
E	1.5 ChitosanTiO_2_Fe_2_O_4_	ChTmF	0.5	0.3
F	1.5 CuTiO_2_Fe_2_O_4_	CTmF	0.5	0.3
G	1.5 ALTiO_2_Fe_2_O_4_	ATmF	0.5	0.3
H	No MNPs (Control)	n/a	0.5	0.3

**Table 3 ijerph-19-09805-t003:** The water quality analysis for BS wastewater.

Setup	COD Removal (%)	Colour Removal (%)	Turbidity Removal (%)
A	92.59	74.86	94.13
B	91.90	61.98	91.94
C	88.07	73.68	84.56
D	91.60	78.95	97.81
E	90.76	68.77	95.90
F	93.70	72.63	95.36
G	79.83	55.61	96.45
H	54.96	45.61	60.79

A (mF), B (NmF), C (CmF), D (TmF), E (ChTmF), F (CTmF), G (ATmF), and H (no MNM).

**Table 4 ijerph-19-09805-t004:** Summary of the kinetic study for bioreactors: A (mF), B (NmF), C (CmF), D (TmF), E (ChTmF), F (CTmF), G (ATmF), and H (no MNM) fitted on first-order and modified Gompertz models.

		Modified Gompertz Model			First-Order Model		
Setup	Measured Yield, (mL/day)	Predicted Yield (mL/day), Y_2_	Y1–Y_2_ (mL/day)	R^2^	Predicted Yield (mL/day), Y_3_	Y_1_–Y_3_ (mL/day)	R^2^
A	1172	1460	288	0.9931	1872	700	0.9688
B	1028	1316	288	0.9943	1956	928	0.9689
C	1004	1174	170	0.9986	1372	368	0.9786
D	848	986	138	0.9952	3476	2658	0.9758
E	804	899	95	0.9960	3387	2583	0.9716
F	710	729	19	0.9854	1162	452	0.9618
G	553	557	4	0.9813	584	31	0.9404

A (mF), B (NmF), C (CmF), D (TmF), E (ChTmF), F (CTmF), G (ATmF), and H (no MNM).

## Data Availability

All data are available in the manuscript.

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
