# Peer review of "Assessment of Magnetic Nanomaterials for Municipality Wastewater Treatment Using Biochemical Methane Potential (BMP) Tests"

_ijerph, 2022, doi:10.3390/ijerph19169805_

Round 1

Author Response

Thanks very much for the critical comments and suggestions for improving the technicality of the manuscript as presented in the revised manuscript.

Reviewer 2 Report

The article "Assessment of magnetic nanomaterials for municipality wastewater treatment via biochemical methane potential (BMP) tests" provides valuable information for applying magnetized nanomaterials for biogas production. The authors must address the following comments-

Page 2, line 46: COVID-19 RNA is not a pharmaceuticals. It should be categorized as biomolecules.

After which treatment (primary/secondary) the wastewater and sludge were collected?

Is table 1 missing the data for activated sludge?

A schematic diagram would be a better fir for figure 1.

Use either colour or color in the article. Do not use both.

Author Response

Thanks very much for the critical comments and suggestions for improving the technicality of the manuscript as presented in the revised manuscript. The table below addresses all the comments
